# MSDZip: Universal Lossless Compression for Multi-source Data via Stepwise-parallel and Learning-based Prediction

Anonymous Author(s)*†

## Abstract

With the rapid development of the Internet, the huge amount of Multi-Source Data (MSD) brings challenges in data sharing and storing. Lossless data compression is the major way to solve those problems. Nowadays, neural-network technologies bring significant advantage in data modeling, making learning-based lossless compressors (LLCs) for multi-source data have emerged continuously. Compared with traditional compressors, the LLCs are more useful to catch complex redundancy patterns in MSD, and thus have great potential in enhancing compression ratio. However, existing LLCs still suffer from unsatisfactory compression ratios and lower throughput. To solve those problems, we propose a novel universal MSD lossless compressor called MSDZip via Stepwise-parallel and learning-based prediction technologies, it introduces two major designs: 1) We propose a Local-Global-Deep Mixing block in the learning-based prediction module to establish dependencies for MSD symbols, where designed Deep Mixing block solves the problem of unstable weights in the perceptual layers caused by cold-start problem to enhance the compression ratio significantly. 2) We design a Stepwise-parallel multi-GPU-accelerated compression strategy to address the compression speed and graphics memory constraints of single GPU in the face of large-scale data. The Stepwise-parallel module passes the source MSD to learning-based prediction model through the data chunking strategy, where the model of the previous chunk is used to guide the compression of the next chunk in parallel. We compare MSDZip with 5 classical learning-based and 6 traditional compressors on 12 well-studied real-world datasets. The experimental results demonstrate that MSDZip optimizes 3.418%~69.874% in terms of compression ratio and 31.171%~495.649% in terms of throughput compared to advanced LLCs. The source code of MSDZip and the linkages of the experimental datasets are available at https://anonymous.4open.science/r/MSDZip-0E4E/.

## CCS Concepts

• **Information systems → Data compression**.

## Keywords

multi-source data, lossless data compression, neural networks, deep learning, parallel computing

**ACM Reference Format:**
Anonymous Author(s). 2025. MSDZip: Universal Lossless Compression for Multi-Source Data via Stepwise-parallel and Learning-based Prediction. In *Proceedings of Make sure to enter the correct conference title from your rights confirmation emai (WWW '25)*. ACM, New York, NY, USA, 9 pages. https://doi.org/XXXXXXX.XXXXXXX

## 1 Introduction

With the rapid development of the worldwide Internet, the volume of Multi-Source Data (MSD), like images, texts, videos and audios, showing explosive growth [11, 47, 48]. As IDC report [42], the global data size is expected to climb to 284 ZB (1 Zettabyte = $2^{70}$ bytes) by 2027. This surge in data volume poses significant pressure and challenges in MSD sharing and storing in the Internet-connected web world. Therefore data compression is critical in the web. For example, compression of CSS and JavaScript files on websites can reduce the file size, thus reducing the number of HTTP requests and transmission time; compression of website data (e.g., PNG [2] lossless format of images), which can improve page loading speed and user experience; and compression of redundant indexes in the server database can reduce the memory consumption and accelerate the query speed at the same time [28].

Universal methods for lossless MSD compression are either traditional or learning-based. Traditional compressors suffer from inferior compression effect, because they fail to fully consider the contextual environment of the redundancy symbols in the to-be-compressed-MSD. Recently, with deeper research into neural-network technologies, an emerging trend is to combine deep learning models with entropy coding algorithms [20, 41] to achieve more efficient lossless compression, such as PAC [33], TRACE [32], OREA [31], DZip [13], lstm-compress [23], et al. The learning-based Lossless Compression method shows significant compression potential on MSD datasets, as their excellent fitting and accurate modeling abilities of to-be-compressd-MSD. However, the existing LLCs still face the following shortcomings.

- **Poor Compression Ratio**. The existing LLCs face poor compression ratio problem, for three reasons: 1) Insufficient context modeling capability for redundant to-be-compressed-MSD; 2) The cold-start problem in the initial compression stage of the dynamic prediction module. 3) Unstable layer contribution in the prediction module.
- **Low Throughput**. The low throughput limits the widespread and use of LLCs, especially in large-scale MSD compression scenarios. For example, in our testing, in one 100 MB MSD dataset, CMIX [22] and NNCP [1] consume up to 67 hours and 18.7 hours in total time, respectively. The throughput of CMIX and NNCP are only 412 and 1483 bytes/second, respectively. We argue for two reasons: 1) Complex deep learning model is not practical for LLC, as the reasoning overhead of the model is expensive. 2) Low

algorithm parallelism degree, especially in the era of multi-GPU.

To address those problems, we propose a novel lossless MSD compressor named MSDZip with the following contributions.

- We analyze the problem of unstable layer contribution in LLCs like OREO [31] and PAC [33], and propose a Feature Extracting Module (FEM) and a Local-Global-Deep three-layer Feature Mixing Module (FMM) to solve them. Here, the FEM is used to extract the features of MSD symbols and FMM includes three crucial components Local Mixing Block (LMB), Global Mixing Block (GMB), and Deep Mixing Block (DMB) for local features mixing, global features mixing, and deep output mixing, respectively.
- We design a Stepwise-parallel strategy to accelerate MSD compression on multi-GPU. It takes advantage of the principle that neighboring regions of the MSD have the same distribution as the next data partition to compress the model obtained from the current data partition. The model obtained is used as a bootstrap model for the compression of the next data partition, alleviating the problem of deteriorating compression ratios due to cold-start problem and parallelism.
- We benchmark our MSDZip and 11 advanced compressors on 12 classical real-world datasets to measure the performance. The experimental results show that our MSDZip outperforms existing compression solutions in terms of compression ratio and throughput.

## 2 Background

Traditional statistical modeling-based algorithms can encode the probability of occurrence of symbols in any given data to get as close as possible to the information entropy [4, 5, 19, 21, 34]. However, it is well known that the probability of the same MSD symbol appearing in different locations depends on its contextual context [13, 23, 33]. Compared with statistical models, learning-based predictive models have more powerful expressive ability to deal with complex patterns, and achieve a large advantage in compression ratio. In this section, we first introduce the compression and decompression process of the LLCs, then give their classification philosophy.

We denote **symbol** as the smallest compression unit, usually one byte, **target symbol** is the symbol to be compressed, **history symbols** denote the symbols adjacent to the target symbol, variable $t$ represents the timestep.

When compressing the target symbol $x_i$ in sequence $S$, the compressor inputs $t$ history symbols $\{x_{i-t}, ..., x_{i-1}\}$ into the predictor to obtain the probability distribution $p(x_i|x_{i-t}, ..., x_{i-1})$ of the $x_i$, and then inputs the probability distribution and $x_i$ into the entropy encoder to compress it into a smaller state $c_i$, where $i = t, ..., |s| - 1$ and $|s|$ is the length of the to-be-compressed-MSD sequence. Meanwhile, the predictor calculates the cross-entropy loss between $p(x_i|x_{i-t}, ..., x_{i-1})$ and $x_i$, and computes the gradient of the loss with respect to the model parameters by the back-propagation algorithm, and then updates the model parameters using gradient descent to improve the model prediction capability.

The decompression process differs from the compression process in that the input to the entropy encoder is a probability distribution of $p(x_i|x_{i-t}, ..., x_{i-1})$ and the compressed state $c_i$, and the output is the original symbol $x_i$, where $i = t, ..., |S| - 1$.

## 3 Related Work

Existing DL-based algorithms are categorized into static, dynamic, and semi-dynamic based on whether or not the compressor updates the model parameters during the compression process [13, 32, 45].

### 3.1 Static

Static compressor uses a probabilistic predictive model to pre-train the to-be-compressed-MSD in multiple epochs before compression and saves the model parameters. At the beginning of compression, after loading the trained model parameters the probabilistic predictor no longer performs backpropagation to update the parameters. DeepZip [12] and DecMac [27] use RNN [44] and its variant LSTM [43] as probabilistic predictive models, respectively. Athough there is no need for backpropagation to update the parameters, multiple epochs of pre-training add additional time overhead. The recently emerging large language model-based compressors LLMZip [51] and LMIC [8] use Chinchilla [16] and LLaMa [50] as predictors, respectively. Despite good compression ability in some datasets they get, possess a large inference cost due to the large model. For example, LLMZip's throughput on the Enwik8 dataset [30] is only 27 bytes/s [45].

### 3.2 Dynamic

Dynamic compressors require no pre-training, it start the model with random parameters, and update the parameters in real time during compression to better fit the data distribution. CMIX [22] mixes the probability distributions of more than two thousand models, including several specialized models, and has very good compression ratio but very slow compression speeds. NNCP [1] and lstm-compress [23] use the LSTM as a probabilistic prediction model, where NNCP also uses the LSTM [43] as a predictor of the probability. NNCP and lstm-compress use LSTM as a probabilistic prediction model, where NNCP also preprocesses the data to extract repeated patterns, mapping the original file to a smaller one, which has a better compression ratio but also a larger time overhead. TRACE [32] uses a variant of Transformer [52], the Performer [3, 25], as a predictor. OREA [31] and PAC [33] use Multi-Layer Perception (MLP) [26] in combination with a sequential mask to achieve compression. sequential masks in combination to achieve compression with less computational cost.

### 3.3 Semi-Dynamic

DZip [13] combines static and dynamic compression algorithms that not only pre-train the data, but also introduce additional models to correct the predicted probability distribution during the actual compression process to achieve better compression ratios.

Existing static and semi-dynamic, despite having better compression ratios overall, require additional time for model pre-training and additional space for storing model parameters for decompression. Dynamic compression algorithms have overall lower time and space overheads, but the presence of later cold-start problems [32] with the model leads to poor compression ratios. Therefore, due to issues such as insufficient modeling capabilities of the model,

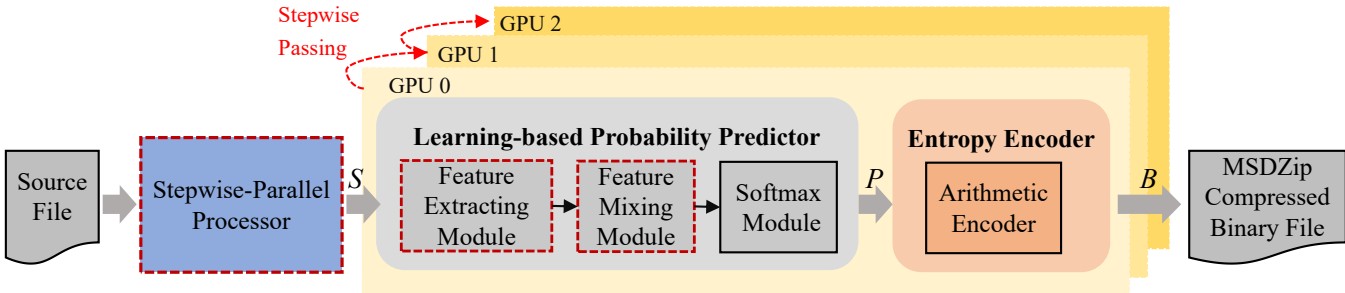

**Figure 1: Compression pipeline of the proposed MSDZip.** $S$, $P$, $B$ **denote the inputted sub-sequence collection, probability distribution collection, and arithmetic encode byte-stream, respectively.**

cold-start problems, and low parallelism design, existing LLCs have significant room for improvement in terms of compression ratio and throughput.

## 4 Method

In this paper, variable $\rho$ represents the parallel number, and variable $d$ denotes the embedding dimension when symbols are input to the model.

### 4.1 Framework

Fig. 1 shows the workflow of the proposed MSDZip, which includes three important components.

- **Stepwise-parallel Processor (SP).** In order to improve the compression speed, SP employs a novel multi-GPU Stepwise-parallel compression strategy. SP splits the read-in byte stream $B$ into $\rho$ equal-length sequences $\{S_i\}_{i=0}^{\rho-1}$ and is responsible for loading the model, inputting the data, and saving the model in a stepped form. Compared with ordinary chunk parallelism, Stepwise-parallel strategy not only reduces the time overhead, but also effectively alleviates the problem of deteriorating compression ratio caused by random parameter startup (cold-start) of the model.
- **Probability Predictor (PP).** PP includes three modules: Feature Extracting Module (FEM), Feature Mixing Module (FMM), and Softmax Module (SM). For each target symbol $x_i$ in sequence $S$, PP inputs the $x_i$ into FEM to obtain the feature vector $h_i$, mixes $h_i$ with the feature vectors $\{h_{i-t}, ..., h_{i-1}\}$ of history symbols vim FMM to establish symbol dependencies and outputs logits $\Lambda_i$, and feeds $\Lambda_i$ into SM to obtain the probability distribution $p(x_i|x_{i-t}, ..., x_{i-1})$ of $x_i$.
- **Entropy Encoder (EE).** EE uses Arithmetic Coding [20, 41] to compress the target symbol $x_i$ into a binary stream based on its probability distribution $p(x_i|x_{i-t}, ..., x_{i-1})$.

### 4.2 Stepwise-parallel Processor (SP)

**Analysis.** LLCs have achieved a large advantage in compression ratio, but the compression speed has always been one of the biggest factors limiting their wide application [45]. With the gradual increase of the data-size, single GPU is overstretched both in terms

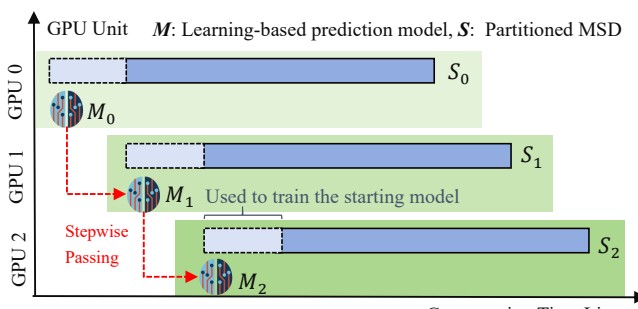

**Figure 2: The schematic diagram of Stepwise-parallel**

of speed and graphics memory, so multi-GPU parallel compression has a wider research prospect [29, 45].

Compared with the static compressor that requires pre-training, the dynamic compressor does not, it starts the model with random parameters to realize the prediction of probability distribution [31–33], and adjusts the parameters in real time during the compression process to converge gradually. As the model is trained while compression, the loss value is gradually decreasing, and the loss value is highly correlated with the information entropy [12, 53]. This means that the compression of those symbols at the beginning of the model startup is not ideal (also called the cold-start problem in [32]. Thus plainly chunking the data and compressing it in parallel at the same time only amplifies this problem.

**Design.** In order to alleviate the problem of poor compression ratio caused by cold-start, we design a Stepwise-parallel Processor, a multi-GPU parallel compression strategy containing three operations: loading, training, and saving model.

Let the initial random parameter model be $M_0$, for any sequence $S_i$ obtained from the segmentation of byte stream $B$, where $i = 0, ..., \rho - 2$. SP first loads the model $M_i$, and then inputs the history and target symbols into PP and EE to realize the compression. The current model state is saved as $M_{i+1}$ after compressing $n$ symbols. While $S_i$ continues to be compressed, $M_{i+1}$ is used as the starting model to start the compression of $S_{i+1}$, and so on, until the last sequence $S_{\rho-1}$ is completely compressed. The whole compression process ends. The Fig. 4 gives an example of Stepwise-parallel by using three GPUs.

In designed SP, all the passed intermediate models are obtained from the previous model training, and the initial $M_0$ can be derived from a fixed random seed, so this parallel compression process does not need to save the model parameters for decompression as in the case of static compressors, but only needs to repeat the above process in reverse.

**Explanation.** The effectiveness of SP mainly utilizes the principle of closer distribution of data in adjacent regions [33]. Taking $S_i$ as an example, to support batch compression, the algorithm sets equally spaced anchor points in $S_i$ and compresses the Target Symbol at the anchor points at the same time, where $i = 0, ..., \rho - 2$. In the next batch, all anchor points are moved backward by one timestep. This means that the model senses the distribution of data in the tail of $S_i$ at the beginning, which in turn is spatially adjacent to the head of $S_{i+1}$. Thus the model trained on $S_i$ can be used to bootstrap the compression against $S_{i+1}$.

### 4.3 Probability Predictor (PP)

**Analysis.** OREO [31] and PAC[33] are the current state-of-the-art LLCs based on MLP [37], and they utilize sequential importance to establish the dependency between symbols. Gradual aggregation of features is achieved by stacking multiple compression modules. In the prevailing view of LLCs, the deeper the neural network, the better the model fits the data, and hence the better the compression [45]. However, as shown in Fig. 3(a), when we used the PAC [33] to test with different layers of compression modules on xml, x-ray, mr and osdb datasets from the Silesia corpus [9], we found that the time overhead keeps increasing as more modules are stacked, but the compression ratio becomes worse instead. Based on this observation, we added a mask with a weight of 1 to the output of each compression module used by PAC, and recorded the change of the mask weight in real time during the compression process. As shown in Fig. 3(b), we have observed that the ranking of the contribution to the final logits are different for compression modules throughout the process. This is because deeper networks may suffer from the problem of vanishing gradients, causing the model to converge more slowly. Especially on small datasets, there may be situations where the model is still converging but the data has already been compressed.

**Design.** In order to improve the performance of the compression model, we propose an improved Learning-based Probability Predictor based on the individual-mix autoregressive compression framework [33], which contains three parts: Feature Extracting Module (FEM), Feature Mixing Module (FMM) and Softmax Module (SM).

The following will introduce each module as an example of compressing the symbol $x_i$ in the sequence $S$, where $i = 0, ..., |S| - 1$.

#### 4.3.1 Feature Extracting Module (FEM).
Feature Extracting Module is used to extract the feature of the symbol $x$.

As shown in Fig. 1, FEM first embeds $x$ to higher dimensions to enhance the representation and capture the semantic relations, yielding $e \in \mathbb{R}^{1 \times d}$. Next, as shown in Eq. 1, the FEM performs linear projection and nonlinear activation [33] on $e$ to obtain the feature

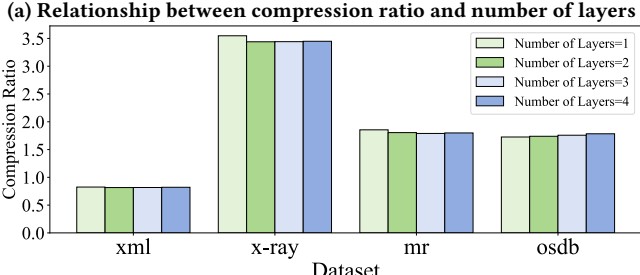

(a) Relationship between compression ratio and number of layers

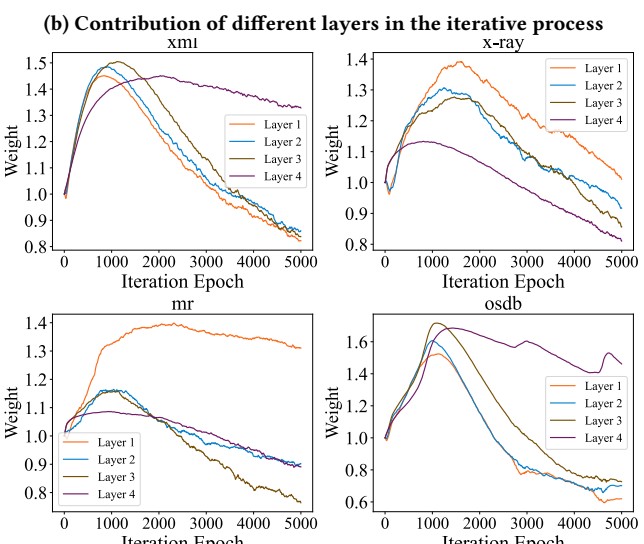

(b) Contribution of different layers in the iterative process

**Figure 3: Influence of layer depth on compression effect**

vector $h \in \mathbb{R}^{1 \times d}$ of $x$.

$$h_\alpha = \delta(e), h_\alpha \in \mathbb{R}^{1 \times d}$$
$$h_\beta = \theta(\delta(h_\alpha W + b)), h_\beta \in \mathbb{R}^{1 \times d}, W \in \mathbb{R}^{d \times d}, b \in \mathbb{R}^{1 \times d} \quad (1)$$
$$h = \frac{h_\alpha + h_\beta}{2}$$

where $\delta$ and $\theta$ are the LayerNorm [24] layer and the activation function GELU [15], respectively, and $W$ and $b$ are the learnable weight and bias, respectively. Learnable weight and bias can enable the establishment of dependencies and ordered importance [31, 33] between symbols.

Based on above feature extraction approach, for the symbol $x_i$ in $S$, where $i = 0, ..., |S| - 1$:

- If $0 \le i < t$, FEM extracts the features $h_i \in \mathbb{R}^{1 \times d}$ of $x_i$ and stores it in Cache, then skips FMM and SM and directly takes the average probability distribution 1/255 of $x_i$ as the output of PP.
- If $t \le i < |S|$, FEM fetches the features $H = \{h_{i-t}, ..., h_{i-1}\}$ of $t$ history symbols from the Cache, and inputs $H$ into FMM for training and prediction to obtain the probability distribution $p(x_i)$ to compress $x_i$. At the same time, FEM extracts the feature $h_i$ of $x_i$ via Eq. 1, and adds it to the tail

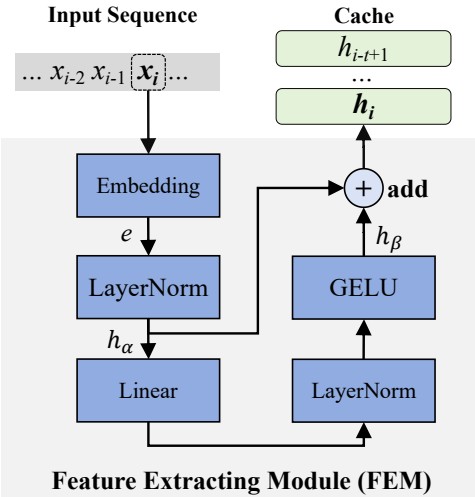

**Figure 4: The schematic diagram of Feature Extracting Module (FEM)**

of Cache while removing the first oldest feature vector $h_{i-t}$ in preparation for compressing the next symbol.

Compared to [12, 13, 23, 32], FEM uses Cache to save features of symbols so that each symbol is processed only once during compression process, greatly reducing computational cost.

*4.3.2 Feature Mixing Module (FMM).* According to the autoregressive compression process in FEM, it can be seen that no dependency has been established between the new input feature vector and the old history features in Cache. For this reason, we design the FMM for mixing the features between the symbols to establish the connection.

FMM consists of several perceptual layers with the same structure, where each layer contains Local Mixing, Global Mixing, and Deep Mixing blocks. FMM receives the feature vectors of $t$ history symbols from FEM to form the initial feature matrix $H^0 \in \mathbb{R}^{t \times d}$. Fig. 5 shows the workflow of each block.

Take the $l$-th Perceptual Layer as an example and define its input as $H^l$, where $l = 0, ..., log_2 t$.

(1) **Local Mixing Block (LMB)**: When $l > 0$, LMB mixes the feature vectors of locally adjacent $2^l$ symbols in $H^l$. LMB first concatenates the local $2^l$ feature vectors to form a new feature vector $h_\alpha^l \in \mathbb{R}^{1 \times (2^l \times d)}$. Then LMB performs linear projection [33] of $h_\alpha^l$ to obtain $h_\beta^l \in \mathbb{R}^{1 \times (2^l \times d)}$ via Eq. 2.

$$h_\beta^l = \theta(\delta(h_\alpha W + b)), W \in \mathbb{R}^{(2^l \times d) \times (2^l \times d)}, b \in \mathbb{R}^{1 \times (2^l \times d)} \quad (2)$$

Finally, the mapped vectors are concatenated and flattened to form $H_L^l \in \mathbb{R}^{t \times d}$.

(2) **Global Mixing Block (GMB)**: GMB mixes the features of all history symbols. GMB first maps $H_L^l$ to higher dimensions to enhance the representation via Eq. 3 to obtain the matrix $H_\alpha^l \in \mathbb{R}^{t \times 2k}$.

$$H_\alpha^l = \theta(H_L^l W_\alpha + b_\alpha), W_\alpha \in \mathbb{R}^{d \times 2k}, b_\alpha \in \mathbb{R}^{1 \times 2k} \quad (3)$$

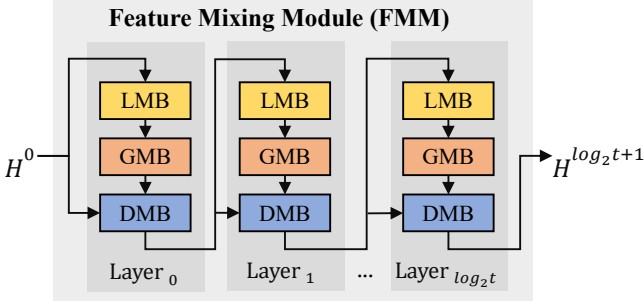

**Figure 5: The schematic diagram of Feature Mixing Module (FMM).** $H^0, H^{log_2 t+1}$ **denote the input and output feature matrices of FMM, respectively. LMB, GMB, DMB represent Local Mixing, Global Mixing, and Deep Mixing block, respectively**

The GMB then employs the spatial gating unit (SGU) [26] to capture more complex patterns in the sequence via Eq.4, and obtain $H_\beta^l \in \mathbb{R}^{t \times k}$.

$$U = H_\alpha^l[:, 0 : k], U \in \mathbb{R}^{t \times k}$$
$$V = H_\alpha^l[:, k : 2k], V \in \mathbb{R}^{t \times k} \quad (4)$$
$$H_\beta^l = U \odot (W_\beta V + b_\beta), W_\beta \in \mathbb{R}^{t \times t}, b_\beta \in \mathbb{R}^{1 \times k}$$

Where the symbol $\odot$ represents the dot product operation. Finally GMB maps $H_\beta^l$ to $H_G^l \in \mathbb{R}^{t \times d}$ via Eq. 5.

$$H_G^l = \theta(\delta(H_\beta^l W_\gamma + b_\gamma)), W_\gamma \in \mathbb{R}^{k \times d}, b_\gamma \in \mathbb{R}^{1 \times d}, \quad (5)$$

(3) **Deep Mixing Block (DMB)**: In order to avoid problems such as gradient explosion or disappearance to improve the compression ratio, DMB mixes $H_G^l$ with $H^l$ via Eq. 6 to generate $H^{l+1} \in \mathbb{R}^{t \times d}$.

$$H^{l+1} = \sigma(\omega_l) \times H_G^l + (1 - \sigma(\omega_l)) \times H^l \quad (6)$$

where $\sigma$ is the activation function Sigmoid and $\omega_l$ is the learnable weight of $l$-th layer, which is initially set to zero.

Finally FMM feeds the output logits $\Lambda = H^{log_t+1}$ to the Softmax Module.

*4.3.3 Softmax Module (SM).* Softmax Module receives the logits $\Lambda$ from the FMM and then flattens it to get $\Lambda_\alpha \in \mathbb{R}^{1 \times (t \times d)}$, then maps $\Lambda_\alpha$ into $|A|$ dimensions to obtain $\Lambda_\beta \in \mathbb{R}^{1 \times |A|} = \{\lambda_0, ..., \lambda_{|A|-1}\}$, where $A$ represents the alphabet. Finally SM gets the probability distribution $p(x_i | x_{i-t}, ..., x_{i-1}) = \{p_j\}_{j=0}^{|A|-1}$ of $x_i$, where $p_j$ is defined as:

$$p_j = Softmax(\lambda_j) = \frac{e^{\lambda_j}}{\sum_{m=0}^{|A|-1} e^{\lambda_m}}, j = 0, ..., |A| - 1 \quad (7)$$

input the $\Lambda$ into the Softmax activation function to obtain the probability distribution $p(x_i | x_{i-t}, ..., x_{i-1})$ of $x_i$.

## 4.4 Entropy Encoder (EE)

Entropy Encoder encodes $x_i$ into a binary stream according to its probability distribution $p(x_i | x_{i-t}, ..., x_{i-1})$ to achieve lossless compression. Common entropy coding algorithms are Arithmetic

Coding [20, 41], Huffman Coding [17] and Asymmetric Numeral Systems [10]. In this paper, we apply the most used Arithmetic Coding in similar studies [7, 12, 13, 31–33] as the entropy encoder because it has the best compression effect.

To easily understand the whole process of MSDZip, we give a detailed description of the compression process as shown in Algorithm 1. Decompression process is similar to compression, except that the input is reversed compared to the compression.

---

**Algorithm 1: Compression Process of MSDZip**

**Input:** input file; timestep $t$; parallel number $\rho$
**Output:** compressed file $\Phi$

1   $B \leftarrow$ Read the input file in byte-stream format;
2   $P \leftarrow$ Initialize the Probability Predictor;
3   $E \leftarrow$ Initialize Arithmetic Encoder;
4   $\{S_i\}_{i=0}^{\rho-1} \leftarrow$ Partition $B$ uniformly into $\rho$ sequences;
5   **function** COMPRESSION($S$, $\varphi$) {
6     Let $S = \{x_i\}_{i=0}^{|S|-1}$ and $\varphi$ are input and output data;
7     **for** $i = 0$ **to** $t - 1$ **do**
8       $p(x_i) \leftarrow$ Get average probability $\frac{1}{256}$ of $x_i$ using $P$;
9       $\varepsilon(x_i) \leftarrow$ Apply $E$ to encode $x_i$ according to $p(x_i)$;
10    **for** $i = t$ **to** $|S| - 1$ **do**
11      $p(x_i|x_{i-t}, ..., x_{i-1}) \leftarrow$ Get probability distribution of $x_i$ using $P$;
12      $\varepsilon(x_i) \leftarrow$ Apply $E$ to encode $x_i$ according to $p(x_i|x_{i-t}, ..., x_{i-1})$;
13      Backpropagate to update $P$ to minimize the loss;
14    Write binary data $\{\varepsilon(x_i)\}_{i=0}^{|S|-1}$ to the file $\varphi$;
15   }
16   $M_0 \leftarrow$ Initialize the model with random parameters;
17   **for** $i = 0$ **to** $\rho - 1$ **Stepwise-parallel do**
18    Load the model $M_i$;
19    COMPRESSION($S_i$, $\varphi_i$);
20    $M_{i+1} \leftarrow$ Save the model at a fixed moment;
21   $\Phi \leftarrow$ Merge all compressed files $\{\varphi_i\}_{i=0}^{\rho-1}$

---

## 5 Results

All experiments were conducted on a server equipped with $4 \times$ Intel Xeon Silver 4310 CPUs (2.10 GHz, 48 cores in total), $4 \times$ NVIDIA GeForce RTX 4090 GPUs (16,384 CUDA cores, 24 GB of GPU memory), and 256 GB of DDR4 RAM. The server runs the Linux operating system Ubuntu 20.04.6 LTS.

### 5.1 Setup

*5.1.1*   ***Datasets***. We used 12 classical well-studied datasets [9, 18, 30, 32, 39, 40, 52, 54–56] of different types to test the performance of MSDZip and similar compressors. The details of the datasets are shown in Table 1.

*5.1.2*   ***Baselines***. We compare MSDZip with 5 advanced open-sourced LLCs lstm-compress [23], DeepZip [12], DZip [13], TRACE [32],

and PAC [33], and 6 classical traditional compressors Gzip [14], PBZIP2 [36], Snzip [49], LZMA2 [35], PPMD [5], and LZ4 [6].

*5.1.3*   ***Metrics***. We measure the compression effectiveness and efficiency of all compressors with the classical metrics, Compression Ratio (*CR*) and Throughput (*THP*), respectively. Among them, compression ratio is defined as [38, 45–47]:

$$CR = \frac{Size_{compressed}}{Size_{source}} \times 8 \; bits/base \tag{8}$$

Here, the $Size_{compressed}$ and $Size_{source}$ represent the size of compressed file and source file, respectively. Smaller values of *CR* indicate the better performance. The *THP* is defined as [45]:

$$THP = \frac{Size_{source}}{CT + DT} \; bytes/second \tag{9}$$

Here, the *CT* and *DT* represent the time costs of compression and decompression, respectively.

*5.1.4*   ***Parameters Setting***. All baselines use the default parameters according there papers and reports [12, 13, 23, 32, 33]. Our proposed MSDZip applies the parameters batchsize, timestep, embedding dimension, and hidden dimension to 512, 16, 16, and 256, respectively, the same as the MLP-based compressor PAC [33]. When executing parallel compression, we used two GPUs, i.e., $\rho = 2$.

### 5.2 Compression Ratio

Table 2 shows the compression ratio of MSDZip with baselines on all datasets. In order to show the advantage of LLC in compression ratio, we have additionally tested the compression ratios of 6 classic traditional algorithms Gzip [14], PBZIP2 [36], Snzip [49], LZMA2 [35], PPMD [5], and LZ4 [6] as a comparison.

From the results, it can be seen that the *CR* of MSDZip on all 12 classical datasets are significantly better than other advanced LLCs and traditional compressors. This is due to the fact that MSDZip employs a proposed Local-Global-Deep Mixing structure in each perceptual layer of the Feature Mixing Module in PP to fully establish the dependencies between symbols, and also introduces the SGU to enhance the information interaction across spatial symbols. We have counted the overall compression ratios of all the compressors and calculated the improvement of MSDZip over baselines via Eq. 10.

$$Improvement = \frac{CR_{baseline} - CR_{ours}}{CR_{baseline}} \times 100\% \tag{10}$$

From the results, we can see that MSDZip has improved by 3.418% ~69.874% in terms of compression ratio as compared to baselines.

Besides, we also found that the lstm-compress [23] performs significantly worse than other compressors on the ImageTest (D8) and DNACorpus (D11) datasets. We believe this is due to gradient explosion or vanishing issues during the training process, which prevents the model from converging well and fitting the data properly. DeepZip [12] seems to exhibit this problem even more severely, as it results in compressed files that are much larger than the source files on datasets such as CLIC (D7), ImageTest (D8), GoogleSpeech (D9), and LJSpeech (D10). On top of its improved version, DZip [13], not only includes pre-training but also introduces an additional model during the compression process to update parameters in real-time, thus avoiding this issue.

**Table 1: Detailed information of all datasets**

| Index | Dataset | Type | Size (bytes) | Description |
|---|---|---|---|---|
| D1 | Enwik8 [30] | text | 100000000 | First $10^8$ bytes of the English Wikipedia dump on 2006 |
| D2 | Text8 [30] | text | 100000000 | First $10^8$ bytes of the English Wikipedia (only text) dump on 2006 |
| D3 | Enwik9 [30] | text | 1000000000 | First $10^9$ bytes of the English Wikipedia dump on 2006 |
| D4 | Book [52] | text | 1000000000 | First $10^9$ bytes of BookCorpus |
| D5 | Silesia [9] | heterogeneous | 211938580 | A heterogeneous corpus of 12 documents with various data types |
| D6 | Backup [32] | heterogeneous | 1000000000 | $10^9$ bytes random extract from the disk backup of TRACE |
| D7 | CLIC [55] | image | 243158876 | Classical image compression benchmark (validation) of the CLIC 2024 |
| D8 | ImageTest [40] | image | 470611702 | A new 8-bit benchmark dataset for image compression evaluation |
| D9 | GoogleSpeech [54] | audio | 327759206 | First 10,000 audio files of the Google Speech Commands Dataset |
| D10 | LJSpeech [18] | audio | 293847664 | First 10,000 audio files of the LJSpeech Dataset |
| D11 | DNACorpus [39] | genome | 685597124 | A corpus of DNA sequences from 15 different species |
| D12 | GenoSeq [56] | genome | 1926041160 | A collection of genomics sequencing dataset with FastQ format |

**Table 2: Compression Ratio (bits/base) of MSDZip and baselines. The Boldface means the best result.**

| Compressor | text | | | | heterogeneous | | image | | audio | | genome | | Overall | Improvement |
|---|---|---|---|---|---|---|---|---|---|---|---|---|---|---|
| | D1 | D2 | D3 | D4 | D5 | D6 | D7 | D8 | D9 | D10 | D11 | D12 | | (%) |
| **Traditional MSD Compressor** | | | | | | | | | | | | | | |
| Gzip | 2.916 | 2.644 | 2.581 | 2.875 | 2.554 | 6.237 | 7.992 | 5.885 | 5.770 | 6.851 | 2.171 | 1.616 | 3.534 | 37.634 |
| PBZIP2 | 2.321 | 2.112 | 2.033 | 2.143 | 2.063 | 5.978 | 7.977 | 4.643 | 4.754 | 5.872 | 2.103 | 1.324 | 3.048 | 27.690 |
| Snzip | 4.475 | 4.303 | 4.020 | 4.678 | 3.829 | 7.100 | 8.001 | 7.326 | 7.387 | 7.982 | 3.682 | 2.676 | 4.800 | 54.083 |
| LZMA2 | 1.989 | 1.858 | 1.718 | 2.027 | 1.839 | 5.084 | 7.956 | 4.786 | 4.717 | 5.850 | 1.800 | 1.198 | 2.799 | 21.258 |
| PPMD | 1.853 | 1.712 | 1.602 | 1.801 | 1.816 | 5.706 | 7.956 | 4.437 | 4.453 | 5.587 | 1.944 | 1.209 | 2.805 | 21.426 |
| LZ4 | 3.359 | 3.117 | 2.980 | 3.263 | 2.921 | 6.653 | 8.000 | 6.658 | 6.514 | 7.728 | 2.734 | 1.977 | 3.985 | 44.693 |
| **Learning-based MSD Compressor** | | | | | | | | | | | | | | |
| lstm-compress | 1.854 | 1.756 | 1.571 | 1.704 | 1.776 | 4.475 | 7.505 | 7.225 | 3.623 | 4.459 | 7.605 | 0.934 | 3.156 | 30.165 |
| DeepZip | 1.952 | 1.806 | 18.027 | 1.681 | 1.870 | 4.340 | 22.204 | 20.272 | 14.398 | 20.833 | 1.863 | 1.018 | 7.316 | 69.874 |
| DZip | 1.875 | 1.757 | 1.539 | 1.591 | 1.716 | 4.236 | 8.021 | 3.595 | 3.805 | 4.921 | 1.799 | 0.908 | 2.366 | 6.847 |
| TRACE | 1.870 | 1.782 | 1.556 | 1.781 | 1.771 | 4.547 | 7.757 | 3.496 | 3.664 | 4.486 | 1.870 | 0.988 | 2.427 | 9.188 |
| PAC | 1.695 | 1.626 | 1.377 | 1.590 | 1.604 | 4.200 | 7.507 | 3.362 | 3.869 | 4.784 | 1.802 | 0.849 | 2.282 | 3.418 |
| MSDZip (Ours) | **1.635** | **1.592** | **1.298** | **1.549** | **1.502** | **4.074** | **7.475** | **3.334** | **3.517** | **4.317** | **1.783** | **0.847** | **2.204** | — |

On the whole, except for DeepZip, which has compression anomalies, the compression ratios of LLCs are significantly better than that of traditional compressors. This is due to the fact that neural network-based prediction models have stronger modeling capabilities compared to traditional statistical methods that can capture more heterogeneous patterns in the data.

## 5.3 Compression & Decompression Throughput

We tested the throughput of MSDZip and baselines on all datasets in the same experimental configurations. The results are shown in Table 3. Table 3 shows that our proposed MSDZip obtains the best *THP* values on all datasets. This is due to the fact that MSDZip uses multiple GPUs for compression, which alleviates the limitations of

a single GPU in terms of speed and graphics memory to achieve compression for large-scale data. At the same time, MSDZip adopts the Stepwise-parallel compression strategy, which greatly improves compression speed while ensuring that the compression ratio does not deteriorate as much as possible.

We also calculate the overall throughput and improvement rate of MSDZip compared to baselines by using Eq. 11.

$$Improvement = \frac{THP_{ours} - THP_{baseline}}{THP_{baseline}} \times 100\% \quad (11)$$

From the results, we see that MSDZip improves 31.171% to 495.649% in throughput compared to baselines.

Table 3: Throughput (bytes/second) of MSDZip and baselines. The Boldface means the best result.

| Compressor | text | | | | heterogeneous | | image | | audio | | genome | | Overall | Improvement |
|---|---|---|---|---|---|---|---|---|---|---|---|---|---|---|
| | D1 | D2 | D3 | D4 | D5 | D6 | D7 | D8 | D9 | D10 | D11 | D12 | | (%) |
| lstm-compress | 2481 | 3529 | 2312 | 2336 | 2096 | 2211 | 2108 | 2024 | 2245 | 1974 | 4027 | 3575 | 2597 | 495.649 |
| DeepZip | 7191 | 5377 | 7465 | 7363 | 7154 | 6186 | 7023 | 6293 | 4613 | 6638 | 10507 | 9262 | 7422 | 108.421 |
| DZip | 4264 | 3820 | 5425 | 3921 | 4207 | 4063 | 4687 | 5052 | 2661 | 4416 | 4483 | 5857 | 4594 | 236.722 |
| TRACE | 12071 | 11328 | 9956 | 12038 | 11175 | 10655 | 10586 | 11230 | 11396 | 10617 | 11416 | 11811 | 11186 | 38.289 |
| PAC | 11733 | 11409 | 12197 | 11671 | 11790 | 11439 | 11018 | 11680 | 11449 | 11046 | 11816 | 12186 | 11793 | 31.171 |
| MSDZip | **15364** | **15446** | **15555** | **15377** | **15604** | **15219** | **14908** | **14985** | **15689** | **15091** | **16057** | **15615** | **15469** | — |

Table 4: The results of ablation study.

| MSDZip- | Component | | | | Compression Ratio | | | | | THP |
|---|---|---|---|---|---|---|---|---|---|---|
| | SGU | DMB | SimP | SP | D1 | D5 | D7 | D9 | D11 | (B/s) |
| A | ✗ | ✗ | ✗ | ✗ | 1.638 | 1.571 | 7.459 | 3.816 | 1.800 | 9389 |
| B | ✓ | ✗ | ✗ | ✗ | 1.605 | 1.539 | 7.453 | 3.801 | 1.795 | 8761 |
| C | ✓ | ✓ | ✗ | ✗ | 1.587 | 1.524 | 7.430 | 3.487 | 1.791 | 8159 |
| D | ✓ | ✓ | ✓ | ✗ | 1.652 | 1.510 | 7.481 | 3.526 | 1.783 | 17367 |
| E | ✓ | ✓ | ✗ | ✓ | 1.635 | 1.502 | 7.475 | 3.517 | 1.783 | 15686 |

**NOTE**: SimP (Simple Parallel) indicates that each data chunk is compressed at the same time with a randomly initialized model.

It is also worth noting that DZip introduces Support Model on the basis of DeepZip, which solves the problem of disappearing or exploding gradients that occurs in DeepZip, and achieves a better compression ratio. However, at the same time, Support Model needs to update the parameters in real time during the compression process, so adds more time overhead. Especially on the DNACorpus (D11) dataset, DZip's throughput is even less than half of DeepZip's. TRACE employs a single-layer Performer as a predictor and a Byte-grouping strategy to reduce the length of sentences for speedup, and thus it ranks third in terms of throughput. PAC has a better throughput than TRACE because it adopts MLP as the predictor with lower computational cost, combined with Ordered Mask to realize lossless compression.

### 5.4 Ablation Study

We performed ablation study on 5 different types of datasets to validate the effectiveness of the additional components used to optimize compression performance. The results are shown in Table 4.

As shown in Table 4, after the introduction of SGU and DMB in turn, the throughput of the compressor is slightly reduced, but the compression ratio is significantly improved. This is because SGU is able to capture the information interactions across spatial symbols when performing the mixing of global features and has a stronger modeling capability. And DMB solves the problem of unstable layer contributions due to cold-start by mixing the outputs between neighboring layers. When using Simple-parallel compression strategy, where multiple data chunks are compressed simultaneously with random parameter initiation, the throughput is significantly

improved, but the compression ratio is slightly degraded by the cold-start problem of the model. When Stepwise-parallel is used instead of Simple-parallel, the compressor achieves a better compression ratio with higher throughput. This is because Stepwise-parallel takes advantage of the principle that data distribution in neighboring regions is more similar to pass the model between data chunks, which alleviates the model cold-start problem to some extent.

In summary, SGU and DMB substantially optimize the compression ratio, while the Stepwise-parallel compression strategy significantly improves the compression speed while maintaining the compression ratio.

## 6 Conclusion

In this paper, we explored two major impacts of the cold-start problem on the compression effect of adaptive learning-based compressors: 1) The cold-start problem leads to different convergence speeds for different perceptual layers, and thus the ranking of the contribution of each perceptual layer changes during the iteration, which leads to different compression effectiveness of symbols at different iteration moments. 2) The parallel acceleration further amplifies the above problem, resulting in poorer compression ratios, despite a substantial increase in throughput. To address them, we proposed the MSDZip with Stepwise-parallel compression strategy and a new learning-based predictor. MSDZip firstly extracts the features of the symbols by Feature Extracting Module, and then fully mixes the features using Local-Global-Deep Mixing Blocks in Feature Mixing Module to sufficiently establish the dependency relationships of symbols. Deep Mixing Block can mix the outputs of upper and lower layers to solve the problem of unstable weight ranking due to cold-start and significantly improve the compression ratio. Stepwise-parallel compression strategy performs contextual modeling of data chunks, and then passes the model among neighboring data chunks with more similar data distributions to mitigate the problem of compression ratio decay due to cold-start while increasing throughput substantially. The experimental results of MSDZip and baselines on 12 well-studied datasets show that our proposed MSDZip has a better compression ratio and higher throughput compared to other advanced compressors.

In the future, we will investigate the impact of data distribution and different neural networks on the performance of the compressor and seek a more efficient and scalable parallel compression strategy.

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
