# OpenReview forum: "MSDZip: Universal Lossless Compression for Multi-source Data via Stepwise-parallel and Learning-based Prediction"
_ACM.org/TheWebConf/2025/Conference — WWW 2025 Poster_

### Official Review · Reviewer_Dtdp · 2024-11-24

**Novelty:** 5
**Technical Quality:** 3

**Review:**

This paper proposed MSDZip, a deep learning-based lossless compression method. The authors introduced the Local-Global-Deep Mixing block and the Stepwise-parallel multi-GPU-accelerated compression strategy, which enables MSDZip to achieve breakthroughs in both compression ratio and compression time.
Pros
1 MSDZip outperforms the baseline in terms of compression ratio and compression time.
2 MSDZip makes full use of multiple GPUs, and solves the problem of unstable weights in the perceptual layers caused by the cold-start problem.
Cons
1 Overall, the authors proposed a number of modules, but the description of the algorithms for each module is not clear enough. For instance, the EntropyEncoder and Algorithm 1 should be explained in more detail.
2 The ablation studies on the various modules proposed by the authors are not sufficient to demonstrate the role of each module, such as the FMM.
3 Compared to traditional algorithms, the authors have applied multiple GPUs to accelerate the compression speed, but they should analyze and compare the consumption of computing resources.
4 In multi-GPU mode, it is not clear that what is the proportion of time that the GPUs truly work simultaneously, and how does the use of multiple GPUs improves the cold start issue for each individual GPU.

**Questions:**

1 How much improvement has been made to the cold start issue by modules such as FEM?
2 Does increasing the dimensionality bring computational complexity?
3 Will the use of multiple GPUs lead to greater resource occupancy?
4 How does the DMB avoid the vanishing and exploding gradient problems?

**Reviewer Confidence:**

2: The reviewer is willing to defend the evaluation, but it is likely that the reviewer did not understand parts of the paper

**Scope:**

3: The work is somewhat relevant to the Web and to the track, and is of narrow interest to a sub-community

---

### Official Review · Reviewer_RvAg · 2024-11-28

**Novelty:** 5
**Technical Quality:** 6

**Review:**

This paper introduces MSDZip, an efficient Multi-Source Data compression system designed to slove the issues of unsatisfactory compression ratios and low throughput in current learning-based lossless compressors. For the first problem, MSDZip designs a Local-Global-Deep network to capture data feature information from three different dimensions, enhancing the model's ability to extract similar information from longer contexts, thereby improving the compression ratios. For the second one, MSDZip adopts a Stepwise-parallel strategy, which improves multi-GPU compression efficiency while reducing the impact of cold-start problems.

**Quality:**  The paper is technically sound, providing a detailed description of the design and principles of MSDZip, which represents the authors' full consideration. It proposes a practical and innovative solution to improve the quality and efficiency of Multi-Source Data compression. The evaluation section presents numerous baselines for comparison with MSDZip, effectively proving its advantages.

**Clarity:** The paper is rigorously argued, with few errors, complete theoretical analysis. However, it lacks necessary explanations in some details (See Question 3 & 4).

**Originality:** MSDZip involves a new neural network model structure and a new parallel solution.

**Significance of this work:** This paper effectively improves the compression ratios and throughput in Multi-Source Data compression, which is of great significance for efficient data storage.

**Pros:**

- The paper is clear in its thinking, complete in content, technically reliable, thoroughly evaluated, and of high quality.

**Cons:**

- The paper has no fatal flaws, only some issues in details (See Questions).

**Fix:**

- In Section 4.3.1, second paragraph: "As shown in Fig. 1, FEM first embeds 𝑥 to higher dimensions to enhance the representation and capture the semantic relations,". This might actually refer to Fig. 4.

**Questions:**

- The compression ratios of the scheme seems to be closely related to the timestep $t$, which affects the number of parameters in the model. However, in the experiment, the time step was set to 16, which does not seem to be a sufficiently large number. This means that the context referenced by the model is only 16 bytes. In Multi-Source Data, many redundant symbols often have a context span that exceeds this value. Please provide more reasons why $t=16$ was chosen.
- It is not clearly stated in the paper whether the runtime environment of baselines is the same as that of MSDZip (which means using two GPUs as well). The computing power of two GPUs is obviously superior to that of a single GPU. How can it be proven that the increase in throughput comes from the design of the model rather than the improvement in computing power? Moreover, from the results of the ablation experiments, Stepwise-parallel did not bring significant effects, and the throughput even decreased significantly (the second GPU is idle in the early stages). Has the trade-off between compression rate and throughput been considered?
- When $0\le i \lt|S|$ , $p(x_i)=1/255$ , which seems to indicate that the probability of  $x_i=0-255$ is the same (but why is it not 256?), but in the Softmax Module, an alphabet is proposed, which seems to indicate that $x_i$ only represents a limited number of symbols （$|A|\le 256$). Is there a contradiction here?
- In section 4.3.2, how is $h_\beta^l\in \mathbb{R}^{1\times(2^l\times d)}$ flattened as $H_L^l\in\mathbb{R}^{t \times d}$ ? The paper seems to lack details on this.

**Reviewer Confidence:**

3: The reviewer is confident but not certain that the evaluation is correct

**Scope:**

4: The work is relevant to the Web and to the track, and is of broad interest to the community

---

### Official Review · Reviewer_xB1Q · 2024-12-02

**Novelty:** 5
**Technical Quality:** 5

**Review:**

This paper presents a novel universal MSD lossless compressor called MSDZip via Stepwise-parallel and learning-based prediction technologies: a Local-Global-Deep Mixing block, and a Stepwise-parallel multi-GPU-accelerated compression strategy to address the issues of unsatisfactory compression ratios and lower throughput. Experimental results show that MSDZip optimizes 3.418%-69.874% in terms of compression ratio and 31.171%-495.649% in terms of throughput compared to advanced LLCs. Generally, this paper is easy to follow and addresses an important problem. However, I have a few concerns.

**Questions:**

First, it is still unclear what are the exact challenges when designing MSDZip?  It seems to me that the solution is easy to figure out?
Second, it seems to be unfair to compare the multi-GPU design with the baselines with only one GPU? I understand that the baselines cannot allow multi-GPU running, but It is better to figure out one more metric to fairly compare.

**Reviewer Confidence:**

3: The reviewer is confident but not certain that the evaluation is correct

**Scope:**

4: The work is relevant to the Web and to the track, and is of broad interest to the community

---

### Official Review · Reviewer_y5T7 · 2024-12-02

**Novelty:** 4
**Technical Quality:** 4

**Review:**

This paper presents a universal Multi-Source Data (MSD) lossless compressor, MSDZip. The proposed method includes (1) a Local-Global-Deep Mixing block to enhance the compression ratio and (2) Stepwise-parallel multi-GPU-accelerated compression strategy to enhance the compression speed. The experiments are conducted on 12 popular datasets, which demonstrated the effectiveness of the proposed approach.

Pros:
1. The paper introduces MSDZip, a novel lossless compression framework designed for multi-source data.
2. This study provides a comprehensive evaluation of the proposed method, utilizing 12 diverse datasets for benchmarking and comparing MSDZip to 11 other compressors, which shows that proposed method can effectively enhance compression ratio and speed.
3. A formal description of the proposed method is provided.

Cons:
1. The selection of arithmetic coding as the entropy encoder should be justified or compared with alternative methods.
2. The paper highlights cold-start as a major issue. However, it lacks a more detailed explanation of how cold-start arises in this specific context and how it impacts the compression performance.
3. The writing of this paper should beimproved. For example, In Section 4.2 (line 346), the sentence "Fig. 4 gives an example of Stepwise-parallel by using three GPUs." should be corrected to: "Fig. 2 gives an example of Stepwise-parallel by using three GPUs." Several minor grammatical errors should be addressed (e.g., Lines 95–98, 191–192, 656–657).

**Questions:**

1. What is the computational cost of running the proposed MSDZip framework, such as GPU memory usage?
2. The proposed method employs an MLP-based architecture. How does the choice of embedding dimensions and number of hidden layers impact the overall performance?

**Reviewer Confidence:**

3: The reviewer is confident but not certain that the evaluation is correct

**Scope:**

3: The work is somewhat relevant to the Web and to the track, and is of narrow interest to a sub-community

---

### Official Review · Reviewer_jTnh · 2024-12-05

**Novelty:** 5
**Technical Quality:** 5

**Review:**

This paper presents MSDZip, a learning-based lossless compression scheme targeting multi-source data. Traditional compressors miss opportunities to identify redundant symbols while existing learning-based compressors are slow. To improve the compression ratio and speed of learning-based compression, MSDZip proposes a feature extraction module and a feature mixing module to fix unstable layer issues in existing approaches, adopts a step-wise parallelization strategy to utilize multiple GPUs, and leverages data locality to mitigate cold start problems. Experimental results show that MSDZip can achieve better compression ratio and speed.

Strengths:
+ The work has a strong motivation. Effectively compressing data is of high significance to the web.
+ The proposed techniques, e.g., stepwise GPU parallelization and feature extraction and mixing in probability prediction, are novel.
+ A variety of datasets are involved in the evaluation.

Weaknesses:
- The improvement over state-of-the-art approaches is not significant. Especially, MSDZip is only 3.4% better than PAC in terms of compression ratio and is only 31% faster (I'm not sure how many GPUs PAC used in the test).
- The proposed techniques do not seem effective at improving compression ratio, as shown in Table 4. I would expect the cold start to have a major impact on the compression ratio, but the compression ratio is nearly the same with and without the solution to the problem. This makes the motivation for these techniques questionable.
- The application scenarios for a multi-GPU compression scheme are not justified. It makes sense in data centers, but in the web, the majority of the devices (phones and PCs) lack one GPU, let alone multiple of them. The real impact of the proposal isn't clear.

**Questions:**

Q1. How many GPUs are used for the LLC baselines in the evaluation?

Q2. What's t in the bullet points at the end of page 4?

In addition, please respond to the weak points above.

**Reviewer Confidence:**

3: The reviewer is confident but not certain that the evaluation is correct

**Scope:**

4: The work is relevant to the Web and to the track, and is of broad interest to the community